# Multimodal Approaches in the Treatment of Chronic Peripheral Neuropathy—Evidence from Germany

**DOI:** 10.3390/ijerph21010066

**Published:** 2024-01-07

**Authors:** Tobias Romeyke, Harald Stummer

**Affiliations:** 1Institute for Management and Economics in Health Care, UMIT—University of Health Sciences, Medical Informatics and Technology, 6060 Hall in Tirol, Austria; harald.stummer@umit-tirol.at; 2Waldhausklinik, Acute Hospital for Internal Medicine, Pain Therapy, Complementary and Individualized Patient Centred Medicine, 86391 Deuringen, Germany; 3University Seeburg Castle, 5201 Seekirchen am Wallersee, Austria

**Keywords:** neuropathy, chronic diseases, multimodal approach, Germany, complex treatment, pain, integrative medicine, quality of life, chronification, multimodal pain therapy, naturopathic complex treatment

## Abstract

Patients with chronic peripheral neuropathy suffer greatly and their quality of life is often restricted. Drug therapy can be accompanied by undesirable side effects and intolerances, or the hoped-for effect does not materialize. Therefore, in addition to drug therapy, attempts are also made to treat the physical symptoms with complementary procedures. In the case of severe forms, the search for a suitable form of therapy is difficult. Complex treatments can be an innovative way to treat peripheral neuropathy. At the same time, several different therapy methods are carried out at high frequency by a specialized treatment team. This study aimed to provide an overview of possible complementary forms of therapy. The focus was on a comparison of two interdisciplinary complex therapies that are used in severe cases in an acute inpatient care setting in Germany. The six dimensions (energy, sleep, pain, physicality, emotional response and social isolation) of the Nottingham Health Profile (NHP) were used to assess quality of life. Both complex treatments (naturopathic complex therapy/multimodal pain therapy) showed a significant reduction in impairment in all dimensions of the NHP. In addition, a multivariate analysis was carried out to take into account several influencing variables at the same time. At the time of admission to the hospital, the degree of chronicity was recorded for each patient. This allowed statements to be made about the effect of the respective therapy depending on the chronification stage of the patient. It has been shown that patients with acutely exacerbated pain with the highest degree of chronicity also benefit from both complex treatments. The naturopathic complex treatment gives the treatment team more options. Aspects such as nutrition, methods from phytotherapy and traditional Chinese medicine can be integrated into inpatient care. Thus, a patient-centered, holistic therapy can take place. However, an interdisciplinary holistic therapy requires more time for both the practitioner and the patient. This should be taken into account in the health systems in the context of the diagnosis related groups.

## 1. Introduction

Peripheral neuropathy is a common neurological disorder resulting from damage to peripheral nerves. Peripheral neuropathy covers diseases with different etiologies. Often there are metabolic, genetic, drug-induced causes or alcohol consumption.

Due to the large extent of its spread, diabetic polyneuropathy plays a special role [1]. The exact pathophysiology of diabetic neuropathy is not well understood, but oxidative stress as well as inflammatory, autoimmune and microvascular mechanisms and hyperglycemia are known influencing factors.

Polyneuropathy can manifest itself through a variety of symptoms. Loss of sensitivity often occurs, with abnormal sensations such as burning, tingling, “pins and needles”, stinging, feelings of swelling, stiffness up to paralysis and disorders of organ function [2]. Polyneuropathy often begins on the feet and legs, and painful muscle cramps in the thighs or calves and dull or stabbing pains in the groin or on the front of the thighs can occur over the course of the disease. The symptoms often worsen at night. Patients are restricted in their mobility, which can mean that they can no longer carry out their daily tasks [3]. Patients in an advanced stage of chronification are restricted in their quality of life [4,5].

First-line therapy with antidepressants and anticonvulsants, but also the use of opiates, are often associated with side effects and show little therapeutic efficiency in many patients [6]. Long-term use of nonsteroidal anti-inflammatory drugs can lead to kidney and gastrointestinal dysfunction [7], long-term use of opiates to constipation, nausea and sleep disorders [8]. Polyneuropathy leads to increased use of healthcare facilities, rising healthcare expenditures and absenteeism from work [9,10].

Studies show that many patients suffering from polyneuropathy often do not receive adequate therapy [11]. This can be due to insufficient knowledge of the patient, the treating physician, ineffective medication, difficult access to effective therapy and other reasons.

In clinical practice, pharmacotherapeutic interventions are often combined with other procedures, since a sufficient effect of monotherapy often cannot be achieved [12]. Early initiation of drug therapy is important. Procedures that are used as a complement to pharmacotherapeutic intervention are usually only evaluated as monomodal therapy in terms of their effectiveness.

For example, the use of acupuncture attempts to stimulate the production of endorphins in the central nervous system [13]. Studies on acupuncture show positive effects on different types of peripheral polyneuropathy, such as those caused by chemotherapy [14,15] or diabetes mellitus [16]. A special type is scalp acupuncture. It is based on the theory of the meridians of traditional Chinese medicine in combination with the functional positioning of the stimulation over the cerebral cortex. Studies show a positive influence on cognitive abilities and an improvement in sensory and motor functions [17]. The mechanism of action of acupuncture cannot be clearly identified, but it has been shown to reduce numbness and pain and improve nerve conduction speed [18]. Techniques from the field of biofeedback and mind–body medicine are also frequently used. They can contribute to reducing pain intensity, pain catastrophization, depression and perceived stress and thus contribute to improving the quality of life [19,20]. Exercise therapy is also an important form of therapy in the treatment of peripheral neuropathy. Exercise therapy aims to improve physical endurance and prevent possible muscle weakness. It can also influence the diabetic metabolic status and blood circulation [21,22]. Massage and foot massage also show a positive influence on the symptoms associated with polyneuropathy [23,24]. Studies on transcutaneous electrostimulation have existed for many years with good treatment results [25]. Wireless transcutaneous electrical nerve stimulation (TENS) can reduce tingling, numbness and pain severity [26]: a systematic review with meta-analysis including 12 studies in patients with diabetic neuropathy showed it produced a greater pain reduction than placebo treatment [27]. Exercises with hands and feet can positively influence the activities of daily living, pain intensity and disease activity [21,28].

Nutrition plays an important role in the prevention and therapy of peripheral neuropathy. It is advisable to consume foods containing vitamins B and E, extracts of medicinal plants, unsaturated fatty acids, as well as foods with phytochemicals. These include turmeric, berries, capers, rocket, buckwheat, dates, etc. They are said to improve the activity of the sirtuin enzymes and have a positive effect on the quality of life of patients with chemotherapy-induced peripheral polyneuropathy [29]. Omega-3 fatty acids also have a preventive character and are important for a healthy cell membrane, blood flow and serve to produce pro-inflammatory cytokines and can improve the macro- and microvascular function of patients [30,31].

The administration of trace elements such as vitamin D can also improve pain symptoms [32]. Vitamin E shows a positive influence on nerve growth factor and an increase in conduction speed [33]. A low-fat vegan diet shows a beneficial effect on patients’ pain intensity and reduction in medication [34,35].

In addition to individual procedures or a combination therapy, multimodal, complex therapies can also be used in severe forms of peripheral neuropathy. So-called complex therapies use high-frequency and indication-related therapy methods in a close temporal context [36]. A multi-professional team must be available, which creates a patient-related therapy plan with therapy goals at the beginning of the treatment.

Measurement of the results of complex treatments in clinical research has so far been seldom represented, as specialized clinics offering this range of services are not very common. Internationally, multimodal pain therapy (MPT) is one of the established, frequently used clinical procedures with increasing scientific evidence [37]. MPT can be used to treat a wide range of diseases. It has been evaluated for chronic back pain, fibromyalgia, facial pain, depression and other clinical pictures [38,39,40].

The therapy density of complex treatment methods compared to the use of monomodal therapy methods is also the subject of scientific studies [41].

It is problematic that there is no uniform definition of the content for multimodal pain therapy [42]. The outcome parameters of the treatment are also not uniformly defined.

As a systematic review by Deckert et al. 2015 shows, the intensity of pain and the disabilities and depression associated with pain are often recorded as outcome parameters [43].

In the German healthcare system, multimodal pain therapy (MPT) is codified as procedure 8-918. The operations and procedures code (OPS) is the official classification for coding operations, procedures and general medical measures. The Federal Institute for Drugs and Medical Devices publishes the OPS on behalf of the German Federal Ministry of Health. It is used in inpatient hospital care. It contains fixed quality indicators for structure, process and result quality.

At least 7 days of interdisciplinary treatment is required for patients with chronic pain conditions. At least two specialist disciplines must be included in the treatment, one of which must be a psychiatric, psychosomatic or psychological–psychotherapeutic discipline. Patients must have at least three of the following characteristics:-Manifest or threatened impairment of quality of life, ability to work and/or regular school attendance;-Failure of previous unimodal pain therapy, pain-related surgery or withdrawal treatment;-Existing medication dependency or misuse;-Pain-maintaining psychiatric comorbidities;-Serious somatic comorbidities.

Treatment management is ensured by a specialist with the additional qualification of special pain therapy. Interdisciplinary diagnostics is carried out by at least two specialist disciplines (a psychiatric, psychosomatic or psychological–psychotherapeutic discipline is mandatory). At least three active therapy methods are used: psychotherapy, physiotherapy, relaxation methods, occupational therapy, medical training therapy, sensorimotor training, workplace or school attendance training, artistic therapy (art and/or music therapy) or other practicing therapies. The therapy sessions last 30 min on average. In group therapy, the group size is limited to a maximum of eight people.

The course of treatment is checked by a standardized therapeutic assessment. There is a daily medical visit or team meeting and an interdisciplinary team meeting once a week.

With the duration of treatment, a different therapy density is specified:-7 to a maximum of 13 treatment days (20 therapy units);-14 to a maximum of 20 treatment days (up to 56 therapy units);-21 treatment days (84 therapy units and more).

Another complex treatment, which is characterized by therapy variety and therapy density, is the naturopathic complex treatment (NCT). It is also codified as a procedure (OPS 8-975). Just like the MPT, it is used in the acute inpatient setting in Germany.

It stipulates that a clinical naturopathic team is led by a specialist with an additional qualification in naturopathic treatment and at least 3 years of experience in the field of traditional naturopathic treatment. In addition to doctors and specialist nursing staff with at least six months of naturopathic experience, the team must belong to at least three of the following professional groups: Physiotherapists/physiotherapists/masseuses/ medical lifeguards/sports teachers, occupational therapists, psychologists, ecotrophologists/diet assistants, art therapists/music therapists.

The clinical naturopathic team treats the patient for at least 120 min of therapy per day. At the beginning of the treatment, a specific naturopathic diagnostic and therapeutic treatment concept must be drawn up. A team meeting must be held at least twice a week. Here, somatic, regulatory therapeutic and social aspects should be discussed with patient-related documentation of the previous treatment results and further treatment goals.

Naturopathic extended care is provided by expert nursing staff.

At least five of the following eight therapy areas must be used: nutritional therapy, hydrotherapy/thermotherapy, other physical methods, phytotherapy, regulatory therapy, movement therapy, draining methods or an additional method (manual therapy, acupuncture/Chinese medicine, homeopathy, neural therapy, artistic therapy (art—and/or music therapy)).

Similar to MPT, the treatment duration specifies a different therapy density:-A minimum of 7 and a maximum of 13 days of treatment and less than 1680 min of treatment;-A minimum of 14 and a maximum of 20 treatment days and less than 2520 treatment minutes, or a minimum of 10 and a maximum of 13 treatment days and at least 1680 treatment minutes;-At least 21 days of treatment or at least 14 days of treatment and at least 2520 min of treatment.

So far, NCT has only been offered in Germany by highly specialized hospitals [44]. MPT is used for various pain disorders, including tumor pain, while NCT can be used not only for all chronic pain disorders but also for various other clinical pictures [45,46,47,48].

Associations, scientific working groups and specialist societies are continually developing the catalogue of operations and procedures. This is done under the aspects of structure, process and result quality.

The aim of this study is to examine the quality of life of patients with peripheral neuropathy who have to be treated in hospital. Furthermore, two inpatient complex treatments are presented and the therapeutic results of both complex treatments are compared. The primary outcome parameter is health-related quality of life. The degree of chronification is determined for each patient according to the Mainz staging according to Gerbershagen in order to be able to make statements about the therapy result depending on the chronification and the complex treatment.

## 2. Methods

We included 254 individuals diagnosed with peripheral polyneuropathy in the controlled study. Their diagnosis had been confirmed by a specialist. In each case, 127 patients received the MPT and the other 127 patients the NCT. The patients did not know which group they belonged to.

The setting was the Acute Hospital for Internal Medicine, Pain Therapy, Complementary and Individualized Patient Centered Medicine, Waldhausklinik Deuringen, Germany.

The application for the implementation of the empirical research project was approved by the Research Committee for Scientific Ethical Questions on 22 March 2023, under the reference number 3223. The study was conducted from 2014 to 2019.

### 2.1. Statistical Data

The Shapiro test was used to check whether the data could be assumed to be normally distributed. Furthermore, there was a descriptive representation of the values separated by group and point in time. Furthermore, the group differences per point in time were examined. Here, either Mann–Whitney tests were used (if the normal distribution was not given) or a *t*-test was calculated. The changes over time were then examined using Wilcoxon tests or t-tests. As an extension of the methodology for answering the research questions, a multivariate analysis was carried out. Repeated measures ANOVA was used. In repeated measures ANOVA, sphericity is a requirement. This is due to the consideration of two points in time. Another prerequisite is that one can assume that the variances of the data are homogenous. This was the case here because the Levene test was not significant (*p* = 0.499). Another requirement for ANOVA is that the residuals are normally distributed. This can be assumed based on the number of cases and the central limit value theorem. The multivariate analysis was intended to test whether there was a group effect with regard to the change in the values and whether the degree of chronification had an influence on the values. It was also checked in each case whether there were any interactions. The visual analogue scale (VAS), Mainz Pain Staging System (MPSS) and Nottingham Health Profile (NHP) are described below.

### 2.2. Visual Analogue Scale (VAS)

The use of the VAS is widespread, and with it, pain measurement can be performed quickly (less than a minute) [49], so its use is economically associated with few resources.

The patient rates their pain on a scale from 0 (=no pain) to 10 (=worst imaginable pain). A particular advantage of the VAS is its high degree of resolution (possibility of the finest grades of judgment) [50]. The reliability and validity of measurements with the VAS has been confirmed [51].

### 2.3. Mainz Pain Staging System (MPSS)

The MPSS measures the severity of pain chronicity. Based on the evaluation, the patients can be divided into three stages of chronification. The time course of the pain, localization of the pain, medication intake behavior and the strain on the healthcare facilities by the patient are recorded [52].

### 2.4. Temporal Aspects of Pain

To determine the frequency of occurrence of pain, a distinction is made between three levels of frequency. Grade 1 records pain that does not occur daily or occurs at most once a day and then returns to zero on a ten-point scale. Grade 2 is pain that occurs several times a day but resolves to zero. Grade 3 records a permanent pain that does not go back to zero. To determine the change in pain intensity, pain fluctuations by two or more scale values (pain scale 0–10) twice or more often a week (grade 1), less than twice a week (grade 2) or a constant pain intensity (grade 3) are differentiated.

To record the duration of pain, hours (grade 1), daily data up to a maximum of one week (grade 2) and weekly data (grade 3) are recorded.

### 2.5. Spatial Aspects of Pain

The patient is asked about the localization of the pain as well as the pain pattern. The specification of a related pain image in one or different regions of the body captures grade 1. Two distinct pain images that can be localized in one or more parts of the body, grade 2. In grade 3, the patient gives more than two distinct pain images.

### 2.6. Medication 

According to the MPSS, pain medication is divided into three groups. Group I includes non-opioid monoanalgesics, group II weakly and strongly effective opioids and group III mixed analgesics, tranquilizers, neuroleptics, cortisone viates, etc.

Grade 1 includes taking medication on fewer than 15 days per month, grade 2 up to a maximum of two group I medications on more than 15 days per month. If the patient takes more than two group I drugs on at least 15 days per month or at least one group II or III drug on at least 15 days per month, grade 3 is affected. To determine the stage, the patient is asked questions about drug withdrawal treatments. Grade 1 is given if there is no withdrawal treatment or a significant dose reduction, grade 2 if a single withdrawal treatment has been carried out, grade 3 if multiple withdrawal treatments have been carried out.

### 2.7. Patient History

The use of the healthcare system by the patient is also the subject of the MPSS. Grade 1 records no change of doctor because of the pain, grade 2 a maximum of three changes and grade 3 more than three. To determine pain-related hospital stays, the patient is asked about the number of inpatient, pain-related hospital treatments. Grade 1 records no or one inpatient pain-related hospital treatment, grade 2 two to three and grade 3 more than three inpatient pain-related hospital stays.

To determine pain-related operations, both outpatient and inpatient surgical interventions are evaluated. Grade 1 does not record any pain-related surgery, grade 2 two to three pain-related surgeries and grade 3 more than three pain-related surgeries. In order to determine the axis stage, a further determination of the degree of the number of pain-related stays in rehabilitation facilities is required. Grade 1 includes none, grade 2 one or two, and grade 3 more than two pain-related rehabilitation measures.

### 2.8. Overall Stage

The overall stage of the patient can be determined by adding the axis stages. Accordingly, for an axis stage of four to six, there is one overall stage of I, from seven to eight II and from nine to twelve an overall stage of III. The validity of the MPSS is assured [53].

In the study, the chronification status was recorded for each patient. The MPSS stages I and II were summarized as low and compared with stage III.

### 2.9. Nottingham Health Profile (NHP)

The Nottingham Health Profile is used to measure health-related quality of life. Quality of life measurement is a central core outcome domain for chronic pain clinical trials [54]. The NHP is a well-validated tool for assessing health-related parametric data. It includes questions on six areas: energy, sleep, pain, physicality, emotional response and social isolation.

All questions have only yes/no answer options and each section score is weighted. The higher the score, the greater the number and severity of problems and limitations. The highest score in a section is 100 [55]. All dimensions of the NHP were recorded at the time of admission to the hospital; i.e., before the start of treatment (T1) and at the time of discharge after the end of therapy (T2).

## 3. Results

Figure 1a records the eight therapy areas of NCP and possible therapy methods that were used. Figure 1b shows the therapeutic areas of the MPT.

The MPT consisted of 76 women and 51 men, and the NCT was also dominated by women with 96 people. The ages in both groups ranged from 39 to 91 years.

The maximum pain intensity 4 weeks prior to inpatient treatment was given as an average of VAS 8.3 (MPT) and 8.51 (NCT).

### 3.1. NHP: Physical Activity (Energy)

With regard to the physical activity dimension, there were no significant differences between the two therapy approaches at both times (Mann–Whitney test, *p* > 0.05) (Table 1).

The Wilcoxon tests used for changes over time within groups were significant for both MPT (*p* = 0.030) and NCT (*p* < 0.001). In both groups there was a significant decrease in the values, i.e., an improvement in physical activity.

Multivariate analysis showed that none of the *p*-values were significant. There were no group differences or interactions. A change over time cannot be assumed here either (Table 1).

### 3.2. NHP: Pain

With regard to the pain dimension, no difference could be determined between MPT and NCT (Mann–Whitney test, *p* > 0.05) (Table 2).

Both groups yielded highly significant *p*-values (Wilcoxon tests *p* < 0.001 MPT and NCT, respectively). It can therefore be assumed that there was a decrease in pain (T1 and T2) in both groups.

The multivariate study showed a significant, moderate temporal effect, *p* = 0.001, p. eta² = 0.101. There was also a significant group difference of MPT and NCT, *p* = 0.049. In the NCT group, there were higher values at admission and slightly lower values at T2 (Table 2).

The plot gives the values regarding the group membership (Figure 2).

### 3.3. NHP: Emotional Reactions

The differences were not significant here (Mann–Whitney test, *p*-values > 0.05). One cannot assume differences between the MPT and NCT groups (Table 3).

Both in the MPT (Wilcoxon test, *p* = 0.002) and in the NCT (Wilcoxon test, *p* < 0.001) there was a significant decrease in the values; i.e., a clear improvement.

The results of the multivariate study showed a significant temporal effect (small effect size), *p* = 0.013, p. eta² = 0.042 (Table 3).

### 3.4. NHP: Sleep

There was also no difference between the two groups in the sleep dimension (Mann–Whitney test *p* > 0.05) (Table 4).

For MPT the *p* value was <.001 and for NCT the *p* value was 0.008 (Wilcoxon test).

There was a significant decrease in the values in both groups from admission to discharge from the hospital.

Repeated measures ANOVA was again calculated to examine the data univariate.

There was a weakly significant interaction between time and MPSS (*p* = 0.05) (Table 4).

While in the MPSS high group at T1 they were at about the same level, the low group at T2 had significantly lower values. This represents the significance in the time-MPSS interaction above (Figure 3).

### 3.5. NHP: Social Isolation

When looking at the dimension of social isolation, there were significant differences between the two groups (Table 5)

There was a significant decrease within the groups for MPT (Wilcoxon test *p* < 0.001) and NCT (Wilcoxon test, *p* = 0.01).

The multivariate analysis showed significant changes over time (decrease in values), *p* < 0.001. However, the effect can only be described as small (p.eta² = 0.041).

Furthermore, there was a difference between the MPT and NCT groups, (*p* < 0.001.) with a small effect (*p* = 0.039) and a significant interaction between time and group membership (*p* = 0.006, *p*.eta²= 0.030).

However, variance homogeneity cannot be assumed because the Levene test was significant (*p* < 0.001), so the results on social isolation should be interpreted with caution (Table 5). The Mann–Whitney and Wilcoxon tests were not affected.

The plot shows a clear decrease in the values overall. The NCT values were smaller at both points in time, the decrease in values was more pronounced in the MPT group—this was also reflected in the significant interaction (*p* = 0.006) (Figure 4).

### 3.6. NHP: Physical Mobility

The Mann–Whitney tests showed no significant differences between the two groups at T1 and T2, since both *p*-values were >0.05 (Table 6).

In both the MPT group (Wilcoxon test, *p* = 0.004) and the NCT group (Wilcoxon test, *p* < 0.001), a decrease in the values over time can be assumed.

The next table again gives the results of the multivariate study.

The multivariate study showed a significant temporal effect, *p* = 0.038 with a small effect size (*p*.eta² = 0.029) (Table 6).

## 4. Discussion

Patient-related outcomes including quality of life are important tools in the assessment of acute exacerbations of chronic diseases and their evaluation of medical treatment outcomes [56,57].

Patients with peripheral neuropathy often have persistent severe pain and their quality of life is restricted, which is also evident from other studies [58]. This is also confirmed by studies comparing the quality of life of patients with other diseases [57]. In the advanced stage, progressive nerve damage is often associated with excruciating pain, limitations in physical functionality, depression and anxiety, so that those affected are often no longer able to carry out their everyday tasks [59].

If the therapy starts too late and is insufficient, for example, in the case of diabetic polyneuropathy, life-threatening foot complications and amputations can result [60]. Therapy should start as early as possible in the disease stage.

Therapy goals must be discussed with the patient before and during the course of therapy. Patients’ expectations that are too high and disappointments should be avoided, as these can result in increased pain.

Important goals of the therapy should be an improvement in sleep, an improvement in the quality of life, the maintenance of social activities and social participation and the maintenance of the ability to work. The administration of medication is first-line therapy, especially as part of a symptom-oriented therapy with anticonvulsants, antidepressants (tricyclics, serotonin-norepinephrine reuptake inhibitors).

Anticonvulsants are suspected to have an effect on alpha2-delta neuronal calcium channels. Larger studies show positive effects on pain reduction in peripheral neuropathy, but also side effects associated with intake, such as dizziness, somnolence, headache, edema and weight gain [61].

Tricyclic antidepressants can be used to treat pain, taking risk factors and side effects into account, since they have an antidepressant and analgesic effect. However, compared to other antidepressants, there are increased cardiovascular risks, as older studies have shown [62,63]. Selective serotonin norepinephrine reuptake inhibitors also show analgesic effects [64] and can be used. In this context, however, an update of the study situation is necessary, since other side effects such as nausea, dizziness, diarrhea, dry mouth and somnolence can often result [65].

Decisive for the decision for an opioid therapy are the individual tolerability and the existence of an adequate liver and/or kidney function of the patient.

Therapy-resistant pain should initially be treated with WHO-II opioids; only if the effect is insufficient can a switch to a WHO-III opiate be made within three weeks [66]. However, all opioid therapies have a similar efficacy and side effect profile. Constipation, dizziness with tiredness, nausea and vomiting, sweating, and states of confusion are often possible, and physical dependence should not be underestimated.

In this study, patients were included in whom the effectiveness of drug therapy was unsatisfactory or side effects occurred and unimodal complementary procedures did not bring sufficient effect. Due to the larger number of subjects included (*N* = 254), medication intake was recorded using Mainz staging according to Gerbershagen (see MPSS methodology, dimension medication intake behavior).

Drug therapy should be supplemented with non-drug therapies in the case of severe courses with chronification. These include unimodal and multimodal therapeutic approaches. Multimodal therapeutic approaches such as the MPT and NCT examined in the study can only be provided in the inpatient setting and should combine monotherapies with proof of effectiveness into a holistic concept.

Integrative multimodal therapeutic approaches offer alternatives to surgical interventions and therapies with opiates [67] and can also help to reduce side effects [68]. Movement therapy, for example physiotherapy on a neurophysiological basis, can be integrated into a treatment concept in order to train the body’s own strength and maintain mobility. Another method is balance training. It is designed to improve coordination and simplify movements by making better use of strength and relieving muscles. Balance training can reduce the patient’s risk of falling and injury [69].

Massage therapy can also be included in the treatment concept as a symptom-reducing therapy [70]. The MPT and NCT integrate movement therapy/exercise therapy methods and physiotherapy.

The transcutaneous electrical nerve stimulation mentioned at the beginning, which aims to reduce pain and improve functional abilities, should be integrated into a non-drug treatment program and continued on an outpatient basis. In the long-term assessment, significant improvements in pain intensity were achieved in patients with diabetic polyneuropathy that had been present for more than 8 years [71]. This form of therapy can also be integrated into NCT and MPT as a physiotherapeutic procedure.

Psychotherapy/behavioral therapy was integrated into both complex treatments and shown to be an important therapeutic pillar in the treatment of chronic subacute pain [72,73].

The NCT also contains therapy areas with lifestyle change measures that take into account somatic and psychosocial aspects in equal measure in the presence of somatic comorbidities (e.g., hypertension, hyperlipidemia, obesity or others). This also includes nutritional therapy.

Other therapy methods (Figure 1a) such as acupuncture are structurally specified only for NCT and their use can be considered as therapy for pain reduction [74].

Local anesthetics are injected for therapeutic purposes in neural therapy, which is also an “other therapy” method of NCT.

What both complex treatments have in common is that they are designed for a longer treatment period (length of stay > 13 days).

The advantage of carrying out a complex therapy with a longer treatment time also lies in the possibility of a more detailed evaluation of the therapy measures used by the interdisciplinary treatment team. If the therapy is successful, methods can also be used whose effectiveness has not yet been sufficiently proven in studies if they prove to alleviate symptoms during the inpatient stay. The NCT provides for an interdisciplinary team meeting twice a week and the MPT once a week to evaluate the patient-related therapy goals. All professional groups of the treatment team must attend this meeting.

When comparing the subjects of both complex treatments, a high average pain intensity in the run-up to the treatment was noticeable, which was also reflected in the “pain” dimension of the NHP on admission to the hospital. This had an impact on the dimension of the “physical mobility” of the NHP, which has also been demonstrated in other studies [75,76].

Differences in the impairment of physical activity (energy) were shown neither in the MPT nor in the NCT. At the beginning of the treatment, the values showed a great impairment, which can be attributed to the underlying disease of diabetes mellitus [77].

Looking at all six dimensions of the NHP showed that both complex treatments led to a significant improvement in health-related quality of life. Differences between MPT and NCT can be seen in the “pain” dimension. Here, the pain intensity in the subjects in the NCT was higher at the beginning of the treatment and lower at the end. This could be due to the fact that the NCT provides methods of pain therapy, such as acupuncture or neural therapy, as “other therapy methods”.

The impairments in the dimension “emotional reactions” improved over time in the MPT and NCT. The decrease in the NCT was more pronounced but not significant.

The dimension pain-related “sleep” was clearly pronounced in patients of both groups at the beginning of the treatment. This was also shown by other studies that have compared patients with diabetic polyneuropathy and other forms of polyneuropathy [57,78].

A group difference between MPT and NCT can be seen in the dimension “social isolation” when being admitted and discharged from the hospital. In the MPT, there was a more pronounced decrease in impairment. This can probably be attributed to the naturopathic extended care that OPS 8-975 structurally specifies. The nursing staff participates in therapy and can thus create a close patient–care relationship. As part of the treatment team, holistic care provides various applications, such as holistic rubs or detoxifying procedures, and therefore carers have a very close relationship with the patient. This includes physical, psychosocial, emotional and spiritual care.

The dimension “physical mobility” decreased significantly between the two points in time, which means that the impairments were significantly reduced in both the MPT and the NCT at the time of discharge, regardless of the degree of chronification.

The dimension pain-related “sleep” showed a greater influence of the therapy in the case of a lower degree of chronification than in highly chronified patients.

Otherwise, the therapy results for all subjects of the other dimensions of the NHP were independent of the degree of chronification. This means that a response to the therapy was possible even with a high, advanced degree of chronification.

Both complex treatments enabled patient-centered, holistic care for patients with peripheral neuropathy. Compared to MPT, NCT emphasizes the variety of therapies, since it integrates aspects such as nutrition or phytotherapy, detoxifying therapy and other procedures; for example, from traditional Chinese medicine. 

However, optimal diabetes control is generally considered an essential first step in the prevention and management of DPN [79].

It is important that clinics continue to specialize and develop special treatment programs in the sense of complex treatments, so that chronic pain patients can seek treatment without long waiting times, so that the disease process does not worsen [80].

Starting therapy early can help slow down the chronification process.

## 5. Conclusions

Therapy attempts with multimodal, naturopathic and holistically oriented treatments, regardless of the chronification status, can contribute to improving the quality of life and maintaining the tasks of everyday life. However, an interdisciplinary holistic therapy requires more time for both the practitioner and the patient. This should be taken into account in the health systems in the context of the diagnosis related groups.

Due to the lack of or insufficient research into multimodal, complementary therapy programs for advanced peripheral neuropathy, long-term studies should be planned and carried out. In the present study, only the period of treatment with the complex treatment was analyzed. Another limitation is that in the majority of patients with peripheral neuropathy, it was triggered by diabetes mellitus.

## Figures and Tables

**Figure 1 ijerph-21-00066-f001:**
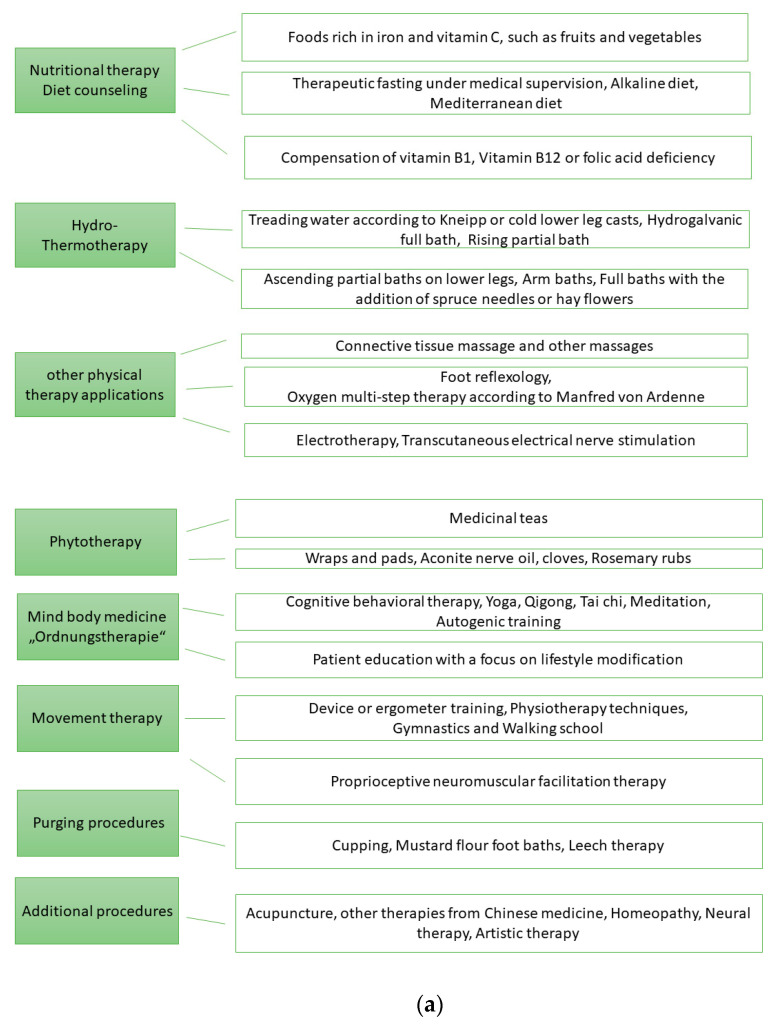
(**a**). Therapy areas of naturopathic complex treatment (NCT). (**b**). Therapy areas of multimodal pain therapy (MPT).

**Figure 2 ijerph-21-00066-f002:**
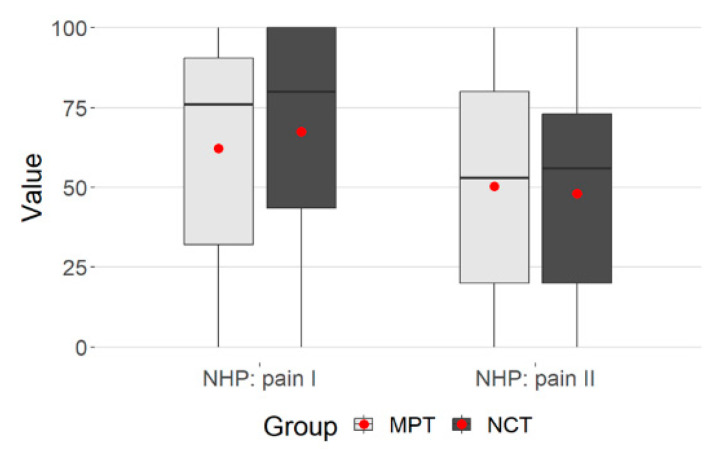
Group membership, NHP: pain.

**Figure 3 ijerph-21-00066-f003:**
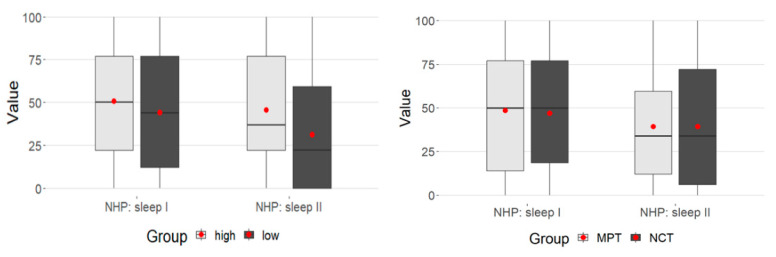
Time-MPSS interaction: sleep.

**Figure 4 ijerph-21-00066-f004:**
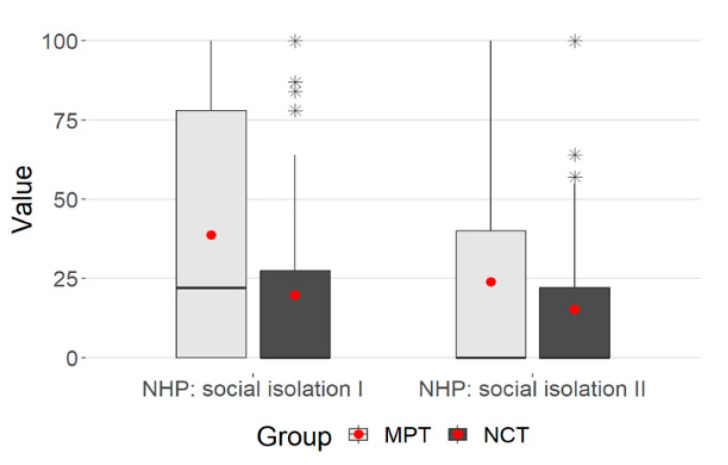
Box plot-NHP: social isolation. (* statistical outlier).

**Table 1 ijerph-21-00066-t001:** NHP: physical activity (energy).

	MPT (*N* = 127)			NCT (*N* = 127)		Total (*N* = 254)		*p* Value	
Item of NHP: physical acitivity T1								0.143	
Mean (SD)	63.685 (39.700)			70.874 (36.786)		67.280 (38.364)			
Item of NHP: physical acitivity T2								0.931	
Mean (SD)	54.465 (37.208)			54.575 (41.051)		54.520 (39.099)			
	b	SE	df	t	*p*-value	p. eta²	CI	low	high
(Intercept)	71.888	7.482	360.48	9.608	0				
time	−5.808	4.492	250	−1.293	0.197	0.07	0.95	0.027	1
MPSSlow	3.134	9.61	360.48	0.326	0.745	0	0.95	0	1
GroupNCT	14.256	9.108	360.48	1.565	0.118	0.007	0.95	0	1
time:MPSSlow	−5.376	5.77	250	−0.932	0.352	0.003	0.95	0	1
time:GroupNCT	−6.275	5.469	250	−1.147	0.252	0.005	0.95	0	1
Levene: F(1, 506) = 1.067, *p* = 0.302									
R² = 0.064									

MPT multimodal pain therapy, NCT naturopathic complex treatment, b regression coefficient, SE standard error, df number of degrees of freedom, t t-distribution, p. eta² effect size, CI confidence interval, MPSS Mainz Pain Staging System, NHP Nottingham Health Profile.

**Table 2 ijerph-21-00066-t002:** NHP: pain.

	MPT (*N* = 127)		NCT (*N* = 127)			Total (*N* = 254)		*p* Value	
Item of NHP: pain T1								0.167	
Mean (SD)	62.220 (35.867)		67.465 (34.753)			64.843 (35.342)			
Item of NHP: pain T2								0.546	
Mean (SD)	50.291 (33.724)		48.110 (30.595)			49.201 (32.153)			
	b	SE	df	t	*p*-value	p. eta²	CI	low	high
(Intercept)	81.675	5.936	391.554	13.76	0				
time	−11.94	3.457	250	−3.454	0.001	0.101	0.95	0.049	1
MPSSlow	−12.881	7.624	391.554	−1.69	0.092	0.007	0.95	0	1
GroupNCT	14.289	7.226	391.554	1.978	0.049	0.01	0.95	0	1
time:MPSSlow	−0.56	4.44	250	−0.126	0.900	0	0.95	0	1
time:GroupNCT	−7.514	4.208	250	−1.785	0.075	0.013	0.95	0	1
Levene: F(1, 506) = 0.835, *p* = 0.361									
R² = 0.129									

MPT multimodal pain therapy, NCT naturopathic complex treatment, b regression coefficient, SE standard error, df number of degrees of freedom, t t-distribution, p. eta² effect size, CI confidence interval, MPSS Mainz Pain Staging System, NHP Nottingham Health Profile.

**Table 3 ijerph-21-00066-t003:** NHP: emotional reactions.

		MPT (*N* = 127)		NCT (*N* = 127)		Total (*N* = 254)		*p* Value	
Item of NHP: emotional reactions T1								0.688	
Mean (SD)		36.638 (34.541)		33.803 (34.181)		35.220 (34.323)			
Item of NHP: emotional reactions T2								0.071	
Mean (SD)		28.654 (36.683)		20.465 (30.955)		24.559 (34.121)			
	b	SE	df	t	*p*-value	p. eta²	CI	low	high
(Intercept)	47.607	6.172	396.426	7.714	0				
time	−8.909	3.577	250	−2.491	0.013	0.042	0.95	0.011	1
MPSSlow	−4.054	7.927	396.426	−0.511	0.609	0.001	0.95	0	1
GroupNCT	3.32	7.514	396.426	0.442	0.659	0	0.95	0	1
time:MPSSlow	1.416	4.594	250	0.308	0.758	0	0.95	0	1
time:GroupNCT	−5.579	4.354	250	−1.281	0.201	0.007	0.95	0	1
Levene: F(1, 506) = 3.547, *p* = 0.06									
R² = 0.037									

MPT multimodal pain therapy, NCT naturopathic complex treatment, b regression coefficient, SE standard error, df number of degrees of freedom, t t-distribution, p. eta² effect size, CI confidence interval, MPSS Mainz Pain Staging System, NHP Nottingham Health Profile.

**Table 4 ijerph-21-00066-t004:** NHP: sleep.

		MPT (*N* = 127)		NCT (*N* = 127)		Total (*N* = 254)		*p* Value	
Item of NHP: sleep T1								0.754	
Mean (SD)		48.575 (36.183)		46.937 (32.189)		47.756 (34.186)			
Item of NHP: sleep T2								0.889	
Mean (SD)		39.291 (33.378)		39.386 (34.969)		39.339 (34.115)			
	b	SE	df	t	*p*-value	p. eta²	CI	low	high
(Intercept)	57.893	5.845	417.437	9.905	0				
time	−6.217	3.312	250	−1.877	0.062	0.032	0.95	0.006	1
MPSSlow	3.873	7.507	417.437	0.516	0.606	0.001	0.95	0	1
GroupNCT	−2.776	7.115	417.437	−0.39	0.697	0	0.95	0	1
time:MPSSlow	−8.374	4.254	250	−1.969	0.050	0.015	0.95	0	1
time:GroupNCT	1.927	4.032	250	0.478	0.633	0.001	0.95	0	1
Levene: F(1, 506) = 0.25, *p* = 0.618									
R² = 0.047									

MPT multimodal pain therapy, NCT naturopathic complex treatment, b regression coefficient, SE standard error, df number of degrees of freedom, t t-distribution, p. eta² effect size, CI confidence interval, MPSS Mainz Pain Staging System, NHP Nottingham Health Profile.

**Table 5 ijerph-21-00066-t005:** NHP: social isolation.

		MPT (*N* = 127)		NCT (*N* = 127)		Total (*N* = 254)		*p* Value	
Item of NHP: social isolation T1								<0.001	
Mean (SD)		38.787 (38.544)		19.709 (29.560)		29.248 (35.587)			
Item of NHP: social isolation T2								0.036	
Mean (SD)		23.890 (35.049)		15.276 (29.853)		19.583 (32.776)			
	b	SE	df	t	*p*-value	p. eta²	CI	low	high
(Intercept)	58.636	5.44	446.067	10.778	0				
time	−16.795	2.977	250	−5.641	0	0.041	0.95	0.01	1
MPSSlow	−6.253	6.988	446.067	−0.895	0.371	0.002	0.95	0	1
GroupNCT	−28.146	6.623	446.067	−4.25	0	0.039	0.95	0.015	1
time:MPSSlow	3.101	3.824	250	0.811	0.418	0.003	0.95	0	1
time:GroupNCT	10.034	3.624	250	2.768	0.006	0.03	0.95	0.005	1
Levene: F(1, 506) = 25.639, *p* = 0									
R² = 0.082									

MPT multimodal pain therapy, NCT naturopathic complex treatment, b regression coefficient, SE standard error, df number of degrees of freedom, t t-distribution, p. eta² effect size, CI confidence interval, MPSS Mainz Pain Staging System, NHP Nottingham Health Profile.

**Table 6 ijerph-21-00066-t006:** NHP: physical mobility.

		MPT (*N* = 127)		NCT (*N* = 127)		Total (*N* = 254)		*p* Value	
Item of NHP: physical mobility T1								0.225	
Mean (SD)		54.181 (26.037)		50.236 (25.661)		52.209 (25.874)			
Item of NHP: physical mobility T2								0.130	
Mean (SD)		47.425 (27.689)		42.441 (24.495)		44.933 (26.208)			
	b	SE	df	t	*p*-value	p. eta²	CI	low	high
(Intercept)	64.936	4.596	370.381	14.128	0				
time	−5.709	2.733	250	−2.089	0.038	0.029	0.95	0.005	1
MPSSlow	0.022	5.904	370.381	0.004	0.997	0	0.95	0	1
GroupNCT	−1.021	5.595	370.381	−0.182	0.855	0	0.95	0	1
time:MPSSlow	−3.809	3.511	250	−1.085	0.279	0.005	0.95	0	1
time:GroupNCT	−1.115	3.328	250	−0.335	0.738	0	0.95	0	1
Levene: F(1, 506) = 0.274, *p* = 0.601									
R² = 0.173									

MPT multimodal pain therapy, NCT naturopathic complex treatment, b regression coefficient, SE standard error, df number of degrees of freedom, t t-distribution, p. eta² effect size, CI confidence interval, MPSS Mainz Pain Staging System, NHP Nottingham Health Profile.

## Data Availability

For data inquiries, please contact the corresponding author.

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
