# Peer review of "Multimodal Approaches in the Treatment of Chronic Peripheral Neuropathy—Evidence from Germany"

_ijerph, 2024, doi:10.3390/ijerph21010066_

Round 1
Reviewer 1 Report
Comments and Suggestions for Authors
Page 3: In the introduction you refer to the selection of at least three active therapy methods. How did you select these methods?
- You have included homeopathy as a therapy, but this supposed therapy, pharmacologically, does not present sufficient evidence to be considered as such.
- You should include in the data shown in the different tables, the values with the sign ±. You must also include the captions describing the abbreviations that appear in your table.
- You should check in table 5 the shading that is coloured in p=0.
- You must check the blanks in the brackets listed before paragraph 3.6
In the conclusions section, the first two paragraphs are referenced. This section is usually not referenced and it seems rather that these paragraphs belong to the discussion. I advise the authors to move these paragraphs to the discussion section.
Author Response
Thank you very much for the review and the valuable comments that improved the manuscript.
Page 3: In the introduction you refer to the selection of at least three active therapy methods. How did you select these methods?
Multimodal pain therapy (MPT) and also naturopathic complex treatment (NCT) is codified as procedure 8-918 and 8-975. The operations and procedures code (OPS) is the official classification for coding operations, procedures and general medical measures. The Federal Institute for Drugs and Medical Devices publishes the OPS on behalf of the German Federal Ministry of Health. It is used in inpatient hospital care. It contains fixed quality indicators for structure, process and result quality.
To clarify this, we have included an explanation in the manuscript.
Associations, scientific working groups and specialist societies are continually developing the catalogue of operations and procedures. This is done under the aspects of structure, process and result quality.
- You have included homeopathy as a therapy, but this supposed therapy, pharmacologically, does not present sufficient evidence to be considered as such.
Homeopathy was included in the complex code 8-975 by medical societies to emphasize the diversity of treatments, although further studies are required to prove its evidence.
- You should include in the data shown in the different tables, the values with the sign ±. You must also include the captions describing the abbreviations that appear in your table.
Yes, thank you. We tried to implement the advice.
- You should check in table 5 the shading that is coloured in p=0.
Yes, thank you for your advice.
- You must check the blanks in the brackets listed before paragraph 3.6
Yes, thank you for your advice.
In the conclusions section, the first two paragraphs are referenced. This section is usually not referenced and it seems rather that these paragraphs belong to the discussion. I advise the authors to move these paragraphs to the discussion section.
Thank you very much, you are right. we have put this part in the top.
Reviewer 2 Report
Comments and Suggestions for Authors
This manuscript by Romeyke and Stummer presents a clinical study on the therapy for chronic pain. The authors compared two complex pain therapies: Multimodal pain therapy (MPT) vs Naturopathic complex treatment (NCT). Based on the Nottingham Health Profile (NHP) for outcome measurement, the authors collected and analyzed data on 254 chronic pain patients that were assigned to either MPT or NCT for comparison of effectiveness of each therapies. The authors found that, in the six out of seven parameters, there is virtually no difference between MPT and NCT in the outcome of pain therapy measured by NHP, except the outcome for social isolation, which showed significant difference between MPT and NCT, with patients in the NCT group scored less social isolation. The study is finely designed and seems well executed with data collection and analyses. The results are very informative and valuable to the pain management research and clinics around the world. A couple of minor issues are here for the author’s considerations to improve the paper.
1. The writing needs to improve to make it easier to read and apprehend the ideas from the authors. For example, in the introduction, discussion, and the conclusion part of the paper, the authors used multiple short paragraphs to describe a similar themed topic, such as the burden of the neuropathy on patients and on society in Introduction. It could be combined together to form a longer paragraph to ease the reading. Similarly, how the therapy for the chronic pain is conducted in the current health care system in the Germany also could be combined into one large paragraph instead of several one-sentence paragraphs.
2. The discussion needs to be more focused on the results. Why the NCT improved social isolation aspect of outcome better than MPT while other parameters were virtually no difference?
3. Fig1 lists the Therapy areas of NCT, which is very good for readers. How about the similar diagram for MPT to aid the readers’ comprehension of the therapy details? Not everyone is familiar with these details.
Comments on the Quality of English Language
Although the paper is okay to read overall, some changes in structure and word choices would greatly improve the readability of the paper.
Author Response
This manuscript by Romeyke and Stummer presents a clinical study on the therapy for chronic pain. The authors compared two complex pain therapies: Multimodal pain therapy (MPT) vs Naturopathic complex treatment (NCT). Based on the Nottingham Health Profile (NHP) for outcome measurement, the authors collected and analyzed data on 254 chronic pain patients that were assigned to either MPT or NCT for comparison of effectiveness of each therapies. The authors found that, in the six out of seven parameters, there is virtually no difference between MPT and NCT in the outcome of pain therapy measured by NHP, except the outcome for social isolation, which showed significant difference between MPT and NCT, with patients in the NCT group scored less social isolation. The study is finely designed and seems well executed with data collection and analyses. The results are very informative and valuable to the pain management research and clinics around the world. A couple of minor issues are here for the author’s considerations to improve the paper.
Thank you very much for your kind words, which motivate us a lot. We would also like to thank you for the valuable review.
- The writing needs to improve to make it easier to read and apprehend the ideas from the authors. For example, in the introduction, discussion, and the conclusion part of the paper, the authors used multiple short paragraphs to describe a similar themed topic, such as the burden of the neuropathy on patients and on society in Introduction. It could be combined together to form a longer paragraph to ease the reading. Similarly, how the therapy for the chronic pain is conducted in the current health care system in the Germany also could be combined into one large paragraph instead of several one-sentence paragraphs.
Yes, thank you. We combined the text and make longer paragraphs for better reading.
- The discussion needs to be more focused on the results. Why the NCT improved social isolation aspect of outcome better than MPT while other parameters were virtually no difference?
This can probably be attributed to the naturopathic extended care that OPS 8-975 structurally specifies. The nursing staff participates in therapy and can thus create a close patient-care relationship. As part of the treatment team, holistic care provides various applications, such as holis-tic rubs or detoxifying procedures, and therefore has a very close relationship with the patient. This includes physical, psychosocial, emotional and spiritual care.
- Fig1 lists the Therapy areas of NCT, which is very good for readers. How about the similar diagram for MPT to aid the readers’ comprehension of the therapy details? Not everyone is familiar with these details.
Yes, thank you very much. We insert Figure 1b with the therapy areas of MPT.